# A Customized Two Photon Fluorescence Imaging Probe Based on 2D scanning MEMS Mirror Including Electrothermal Two-Level-Ladder Dual S-Shaped Actuators

**DOI:** 10.3390/mi11070704

**Published:** 2020-07-21

**Authors:** Hussein Mehidine, Min Li, Jean-Francois Lendresse, Francoise Bouvet, Huikai Xie, Darine Abi Haidar

**Affiliations:** 1Université Paris-Saclay, CNRS/IN2P3, IJCLab, 91405 Orsay, France; mehidine@imnc.in2p3.fr (H.M.); jf.lendresse@hotmail.fr (J.-F.L.); francoise.bouvet@ijclab.in2p3.fr (F.B.); 2Université de Paris, IJCLab, 91405 Orsay, France; 3Suzhou Institute of Biomedical Engineering and Technology (SIBET) Chinese Academy of Sciences (CAS), Suzhou 215163, China; limin@sibet.ac.cn; 4School of Information and Electronics Technology, Beijing Institute of Technology, Beijing 100081, China; hk.xie@ieee.org

**Keywords:** micro-electro-mechanical system (MEMS) mirror, two-photon imaging, electrothermal actuators, 2D scanning probe

## Abstract

We report the design and characterization of a two-photon fluorescence imaging miniature probe. This customized two-axis scanning probe is dedicated for intraoperative two-photon fluorescence imaging endomicroscopic use and is based on a micro-electro-mechanical system (MEMS) mirror with a high reflectivity plate and two-level-ladder double S-shaped electrothermal bimorph actuators. The fully assembled probe has a total outer diameter of 4 mm including all elements. With a two-lens configuration and a small aperture MEMS mirror, this probe can generate a large optical scan angle of 24° with 4 V drive voltage and can achieve a 450 µm FOV with a 2-fps frame rate. A uniform Pixel Dwell Time and a stable scanning speed along a raster pattern were demonstrated while a 57-fs pulse duration of the excitation beam was measured at the exit of the probe head. This miniature imaging probe will be coupled to a two-photon fluorescence endomicroscope oriented towards clinical use.

## 1. Introduction

Today, total resection is still the main therapy to deal with brain tumors. Surgery is considered critical for brain tumor management, where the main challenge is to accurately identify tumor margins to improve its resection outcomes [1]. Tumors boundaries often have similar visual appearance with the surrounding healthy areas, which makes the surgeon unable to delineate these margins correctly and thus results in subtotal removal and tumor recurrence [1]. To obtain more information about these margins and to confirm the success of the surgery, biopsy samples are extracted for histological analysis, which involves Hematoxylin and Eosin (H&E) staining. The results from this post-surgery analysis are only provided several days after to form the absolute diagnosis. This is apart from the fact that if the results show that tumor cells were not well extracted and a second intervention is needed, the final diagnosis relies on human judgement based on expertise and experience of the pathologist. Several methods have been transferred to the operating room to resolve the inherent limitation of biopsy-based histology such as functional brain mapping [2] and intra-MRI [3]. However, these intraoperative modalities have not yet reached the reliability and the precision of the standard H&E post-surgery analysis. However, optical imaging is an emerging field that permits high-resolution and cross-sectional imaging of biological tissues such as Optical Coherence Tomography (OCT) imaging [4] and Two-Photon Fluorescence Microscopy (TPFM) [5]. Both OCT and TPFM can be used endoscopically to perform intraoperative imaging for cancer detection without the use of exogenous dyes. TPFM has been shown to quantitatively and qualitatively discriminate healthy brain tissues from tumoral ones at the subcellular level, therefore competing with H&E stained images [5,6]. The label free and superior spatial resolution of TPFM renders it very useful as a non-invasive diagnosis technique for biomedical imaging applications.

To this end and to address these issues, our team is developing a Two-Photon Fluorescence (TPF) endomicroscope, dedicated to intraoperative label-free imaging for brain tumor tissues [7,8]. This tool is addressed in order to improve the surgical act by helping the neurosurgeon to obtain a real-time, fast and reliable diagnosis response of the examined tissues.

In parallel, and over several years, we have been establishing an optical tissue database which aims to extract and to specify each brain tissue type with its specific optical signature. These specific signatures are derived from the multimodal analysis of the endogenous fluorescence emission of several fluorophores presented in brain cells. In our past studies, we managed to discriminate, with high specificity and sensitivity, healthy human brain tissues, from secondary and primary brain tumors [5], low and high grade glioma [6], and grade I and grade II meningioma [9]. This database will be associated to the endomicroscope as a standard reference to establish a real time diagnosis during intraoperative imaging. In this context, our endomicroscopic tool is able to efficiently excite four main endogenous fluorophores, that their fluorescence signal is analyzed in the tissue database: Nicotinamide adenine dinucleotide (NADH), Flavins (FAD), Lipopigments and Porphyrins, as well as detecting Second Harmonic Generation (SHG) from non-centrosymmetric molecules such as collagen structures presented in brain tissues.

However, transition from benchtop microscopy to endomicroscopy is challenging. Most of the TPFM scanning systems have been developed using free-space optics and benchtop microscopes setups while in vivo imaging and clinical applications impose the use of a fiber-optic endoscope, where the laser beam is delivered through a flexible fiber and image acquisition can be performed using a miniature probe. Choosing the proper fiber is essential to ensure the delivery of femtosecond laser pulses to the imaged specimen. This transition also requires the miniaturizing of the distal optics and integration of the scanning device into medical instruments such as trocars that are used to perform tissue biopsy.

Through time, several miniaturized scanning techniques have been developed to acquire images through TPFM- and OCT-based probes. The two main, widely used scanning techniques are scanning a fiber or scanning an optical beam in free space. The first one consists of introducing the endoscopic fiber into a piezoelectric (PZT) ceramic tube in order to induce resonance within the fiber and perform scanning patterns [10,11]. This technique is widely used in TPFM probes due to its low cost and fast scanning speed. However, it suffers from different mechanical constraints that affect its linear displacement [12] and affect its control to acquire homogenous and repetitive scanning patterns and limit its Field Of View (FOV) [11,13].

The second scanning method is scanning an optical beam using a Micro-Electro-Mechanical System (MEMS) mirror. It is a recent technology widely used for OCT–based endoscopes [14], but not frequently used for TPFM probes due to the high-development cost. The main advantages of this technology reside in its small size, large achievable scanning FOV and stable scanning pattern [14], all of which allow for high scanning frame rates and ease of control. Several groups around the world have designed scanning systems based on MEMS mirrors dedicated mostly for OCT endoscopic imaging [14,15,16]. The design of the probe, lens configuration, total outer diameter of the probe, mirror size and actuation type were the most development challenges faced over time [17,18]. In fact, clinical in vivo imaging and real-time biopsy requires the use of a small diameter probe, with fast acquisition and high resolution. The majority of developed TPFM or OCT probes based on MEMS mirror scanners either have a large diameter due to the use of a large aperture mirror [19], or lack resolution due the non-adapted lens configuration or lack stability and high performance due to the use of a non-adapted actuation type [19].

In our previous works, we managed to overcome the first challenge, i.e., the ultrashort pulse delivery was realized through a homemade micro-structured double-clad photonic crystal fiber (DC-PCF), specially designed for non-linear endomicroscopy [7]. We managed to generate an 810 nm laser beam with a 40-fs pulse duration at the exit of a 5-m long fiber. Briefly, it was composed of a 6.4 µm diameter single mode central core to ensure the excitation with a numerical aperture of 0.097 at 800 nm. The central core was surrounded by an air/silica micro-structured region with a diameter of 40 µm to separate it from the collecting inner cladding. Efficient collection of the fluorescence signals was ensured through this latter waveguide with a numerical aperture of 0.27.

Once the pulse delivery was addressed, our second challenge was to design and manufacture the best performing scanning probe. Our goal is to obtain an intra-operative imaging tool able to perform fast stable scanning acquisition, acquire high resolution images with a large FOV and a miniature probe that is small enough to fit in a surgical tool (surgeon trocar) to perform real-time optical biopsy examination.

In this paper, we report a homemade TPF probe based on a 2D electrothermal MEMS mirror adapted for in vivo clinical imaging in a clear and a defined medical needs context. This customized probe combines a two-lens configuration design with a new actuators shape consisting of two-level-ladder double S-shaped electrothermal bimorph actuators and employs a small aperture mirror. We discussed the choice of the presented lens configuration, probe design, actuation type and actuators architecture. Critical issues such as the mechanical characteristics of the MEMS mirror actuators and the efficiencies of the laser pulse delivery and fluorescence collection were addressed.

## 2. Materials and Methods

### 2.1. Probe Design

The first step in the fabrication of an intraoperative endoscopic probe is to choose the best design that can ensure a high imaging resolution, large imaging FOV and high packaging flexibility within the smallest possible outer diameter.

In their study, S. Tang et al. [18] presented three different side-viewing imaging incorporating a different lens configuration each. They compared their advantages and disadvantages in order to perform multiphoton endoscopy imaging. Their probes implement an electrostatic MEMS mirror with a large diameter (2 mm), able to achieve a 20° as optical scan angle over 90 V as driving voltage. The three different configurations were tested and used to image 20 µm beads in order to compare the image generated from each design.

The first one presented is a common lens configuration widely used in the reported OCT probes designs [15,16], and consists of a single focusing GRIN lens placed between the endoscopic fibre and the MEMS mirror. This configuration is the simplest one that could be realized for an endoscopic probe design. It has the advantage of an easy alignment and assembly packaging as well as size efficiency [18]. The achievable imaging FOV of their tested probe was 200 µm while the resolution of the 20 µm beads was ≈5 µm. However, this configuration imposes the use of a long working distance GRIN lens in order to let the beam reach the sample, which leads to a poor image resolution.

The second design consists of putting the GRIN focusing lens after the MEMS mirror, very close to the sample, which requires the use of a short working distance focusing lens. Therefore, this design is difficult to optimize and suffers from a small achievable FOV (100 µm). In fact, and by using only one lens, the path from the endoscopic fiber to the lens and from the lens to the sample is restricted, which limits the flexibility to adjust those distances, so to fit in the MEMS mirror. It also limits the imaging FOV and imposes the use of a large aperture MEMS mirror in this case, which increases the total outer diameter of the probe. However, this design overcomes the resolution problem of the first design due to the short working distance of the lens, where the resolution of the obtained image of the 20 µm beads was ~2 µm.

The third design consists of a two-lens configuration, where a collimation lens was placed between the fiber and the MEMS mirror and a focusing lens was placed after the mirror close to the sample. This design combines the advantages of the previous design a high FOV, image resolution and flexible in packaging. The path between the collimation lens and the focusing lens can be easily varied to fit in the MEMS mirror without changing the beam property due the collimated beam. Therefore, the obtained image FOV was 200 µm with a 2 µm resolution for the 20 µm beads image.

As the two-lens configuration has the best combination of FOV, resolution and packaging flexibility, we decided to adopt it as the design for our TPF handled probe. We tried also to choose the most miniature possible optical elements while preserving a high performance.

As the collimation lens, an achromatic doublet lens (NT65-564, Edmund Optics, Barrington, NJ, USA) with 1 mm as diameter and 1.5 mm as focal length was used. As the focusing lens (354140-B, Thorlabs, Newton, NJ, USA), an aspheric lens with a large diameter (2.4 mm), a high Numerical Aperture (NA = 0.58) and a short working distance (0.9 mm) was used. Described in the next subsection, the used mirror was a small aperture electrothermal MEMS mirror with a mirror plate size of 0.52 mm × 0.52 mm. Afterwards, Zemax simulations were performed for the chosen design. Through these simulations, we managed to reach a 560 µm FOV with only ±10° as Optical Scan Angle (OSA). Figure 1a shows the distances set between the collimating lens, the MEMS mirror and the focusing lens. The footprint diagram of the tracked beam at the exit of the focusing lens shows that a 560 µm FOV could be reached with a mechanical scan angle of ±5° (Figure 1b).

### 2.2. MEMS Mirror and Probe Assembly

Moving to the MEMS mirror actuation, several types of MEMS mirrors exist, based on electrostatic [20,21], piezoelectric [22], electromagnetic [23] or electrothermal actuation [16]. Electrostatic MEMS mirrors are particularly known for their low power consumption and the high operation frequency (~10 kHz) which enables very fast scanning acquisitions. However, they typically need high driving voltage (~100 V), their scanning ranges are limited compared to the other types (±10°) and their linearity control is complex [14]. Electromagnetic MEMS mirrors have low driving voltages (~3 V) and large scanning ranges (±20°), but they suffer from a low fill factor (5%) and large packaging size due to the need of packaging permanent magnets [14]. Electrothermal MEMS mirrors have several main advantages: high fill factor (25%), low drive voltage (<5 V) and larger scanning range (up to ±30°) compared to the other aforementioned types [14]. In the electrothermal MEMS mirrors family, there are several actuator shapes, such as Z-shape, U-shape [24] and S-shape actuators [15].

A number of electrothermal MEMS mirror-based probes have been reported in the literature, with most of them deploying single-level S-shaped bimorph actuators [15,16,25]. This type (single level architecture) can achieve ~ ±12° as optical scan range [15] using a low voltage with a fast response time (~4 ms) [25]. However, it suffers from a short stable frequency range before reaching the resonance regime [15], while the use of a high voltage (>5 V) may damage the actuators.

Stacking more levels in the actuators architecture has been explored to achieve large piston displacements and to increase the frequency stability and its flat range before resonance [26,27].

S. Luo et al. [26] used an electrothermal MEMS mirror which employs three levels of dual S-shaped bimorph actuators to develop an OCT imaging probe including a C-lens instead of a GRIN lens in a single-lens configuration for the probe design. It has ±17.5° as an optical scan range at only 5 V, with the ability to use a higher voltage to achieve a larger FOV without damaging the actuators.

J. Chai et al. [27] reported a MEMS mirror with three levels of dual S-shaped bimorph actuators dedicated to develop a Fourier transform spectrometer and obtained a more stable scanning with large displacement, but the response time of this shape of actuators was significantly high (28 ms) and needed a high driving voltage (up to 10 V) to achieve a large scanning range [27].

Hence, the use of a new actuator shape, based on a symmetrical two-level ladder of dual S-shaped bimorph actuators (Figure 2a), is adopted. This allows us to keep the low response time of the actuators and to gain more stability and control of the scanning process following a raster scanning pattern.

Table 1 shows the differences among single-level, double-level and three-level MEMS mirrors with the same Al/SiO2 bimorph materials as used for our MEMS mirror actuators. The values concerning the double-level MEMS mirror consist of the values obtained in this work and presented in the results section.

A side-view scanning electron microscopic (SEM) image of the MEMS mirror (WiO Technologies, Wuxi, China) is shown in Figure 2a, where the central mirror plate was suspended with four symmetrical two-level-ladder, double S-shaped bimorph electrothermal actuators over a silicon substrate. An Aluminum (Al) layer covered by a SiO_2_ layer from each side composes each actuator. Figure 2b shows a top-view picture of the MEMS mirror whose footprint was 1.3 mm × 1.5 mm. The surface of the mirror plate was coated with a layer of aluminum (thickness: ~0.3 μm). The end result was an assembled probe with a 4 mm outer diameter, as shown in Figure 2c. The pictures of the mechanical components are shown in Figure 2d.

### 2.3. Characterization Setup

Two different setups were used to measure the optical and mechanical properties of the MEMS scanning mirror. The first one was a reflectivity measurement setup. It consisted of a Xenon light source (HPX-2000, Mikropack, Ostfildern, Germany) coupled with a monochromator (MonoScan 2000, Mikropack, Ostfildern, Germany) delivering a monochromatic beam going from 400 nm to 800 nm with a 1 nm step interval. The outgoing beam was converged to the mirror plate using an objective in order to focus the beam into a 320 µm diameter focal point, smaller than the mirror plate surface, to ensure a total reflection through the mirror. The optical power was measured, using a photodiode sensor (PD300, Ophir photonics, Ophir Spiricon Europe GmbH, Darmstadt, Germany) with a spectral detection range from 350 nm to 1100 nm, before and after the mirror, resulting in the reflectivity at various wavelengths.

To measure the scanning properties of the mirror actuators, the second characterization setup, presented in Figure 3, was used. Each pair of the actuators generates the scanning along a single axis, i.e., the actuators 1 and 4 responsible for the *Y*-axis scanning and the actuators 2 and 3 for the *X*-axis scanning. A laser beam, going out from a laser diode, was directed into the mirror plate and reflected toward a charge-coupled device (CCD) camera (Fastcam Mini UX 50, Photron, Photron Deutschland GmbH, Reutlingen, Germany) to track the laser beam’s spatial displace ment. This camera could reach a recording speed of 160,000 frames per second (fps) and was equipped with a 1.3-Megapixel CMOS Sensor with a pixel size of 10 µm. An AC signal generator (AFG3022C, Tektronix, Beaverton, OR, USA) was used to provide voltage signals to the mirror actuators. This setup was used to measure the quasi-static angular scan characteristics and frequency response of the MEMS mirror.

## 3. Results

Characterization started by investigating the spectral reflectivity of the mirror plate among a polychromatic signal wavelength range. Quantifying this parameter is required in order to calibrate the detected fluorescence emission spectrum so that accurate spectral fitting can be performed. The measured reflectivity in the wavelength range from 400 nm to 800 nm is plotted in Figure 4. The reflectivity at 800 nm, corresponding to the excitation laser, was found to be around 84%. In the collection mode, a similar reflectivity was found at 550 nm, which corresponds to the FAD emission peak [6], while it was found to be lower at 405 nm (74%), which corresponds to the SHG peak when excited at 810 nm. The wavelength range of the TPF signal excited using a 800 nm laser ranges from 400 to 700 nm.

OSA measurements were performed using several voltages ranging from 0 to 4 V. The measured OSA values of the four actuators were plotted in Figure 5. The four obtained curves, corresponding each to a single actuator, showed a quasi-linearity variation starting from 0.5 V. This linearity was investigated by fitting each curve (starting from 0.5 V) to a polynomial fit curve (y = a*x + b) where the fit equations were written and plotted in black (Figure 5). The corresponding R-square values were obtained higher than 0.99 for all fitted curves (0.9927, 0.996, 0.9969 and 0.9902 for actuator 1, 2, 3 and 4 respectively). Indeed, the obtained equations show the excellent similarity, in terms of OSA variation, between the actuators of each scanning axis where the obtained slopes of actuator 1 and 4 were very close and symmetrical (6.909 and −6.839 respectively) as well as those of actuator 2 and 3 (4.03 and −4.01 respectively).

It was observed also that the *Y*-axis actuators had maximum OSAs (24.78° and 24.67° for actuators 1 and 4 respectively at 4V) larger than the *X*-axis actuators (14.31° and 14.49° actuators 2 and 3 respectively at 4V). This difference is occurred due to the non-uniformity in the thickness of the deposited Al/SiO2 layers in the actuators of each scanning axis. This non-uniformity led to an increase in the thermal resistance of the *X*-axis actuators, which directly depends on the thicknesses of these layers [28]. Therefore, the vertical displacement of the *X*-axis actuators was decreased.

However, according to the simulations shown in Figure 1b, a 560-µm FOV only required a ±5° mechanical scan angle, corresponding to ±10° optical scan angle. This angle, and according to the fit equations, could be reached using 2 V as driving voltage for *Y*-axis scanning actuators (1 and 4) and at 2.95 V for *X*-axis scanning actuators (2 and 3).

The obtained linearity of the OSA with the driving voltage ensures a constant Pixel Dwell Time (PDT) along a single scan axis. PDT is defined as the time that the laser beam rests on a single pixel and illuminates it. It characterizes the scanning speed of any scanning system. Therefore, the higher the PDT is, the higher the number of photons collected per pixel and vice versa. The PDTs for Y and X scanning axes were extracted using the videos of the displacement of the reflected laser beam recorded with the CCD camera (Figure 3). The MEMS actuators were driven by an AC generator, which provided a 2 V amplitude triangular wave-form signal with a 10 Hz frequency, while the CCD camera frame rate was set at 20,000 fps.

The obtained results, plotted in Figure 6, show that a quasi-constant PDT was obtained for both X and Y scanning axes along a straight line. The standard deviation of each distribution of the values did not exceed 10% (100 µs) around a calculated mean value of 1000 µs. We can conclude that these two pairs of actuators are able to provide a stable scanning pattern with a uniform scanning speed along all scanning lines. Looking at the borders of these two curves, we can observe that the PDT was high at the beginning and started to decrease to become quasi-constant at the rest of the scanning line. This is due to the fact that the actuators have certain thermal response times before moving when they are driven by a voltage. For that, we measured the response time of each actuator using a 1-V amplitude square-wave signal for each actuator. The AC generator and the CCD camera were triggered together in a way that the CCD camera started recording at the same time as the AC generator provided its square signal. Figure 7a–d shows the obtained rise response times for the four actuators. The orange curve in each figure represents the assumed displacement of the laser beam for an ideal response time. The *Y*-axis represents the position of the laser spot in the CCD camera sensor referred by its pixel index position. The displacement of the laser spot goes to its final position with an exponential variation. For actuators 1 and 4, the 10–90% rise times were 4.08 ms and 4.10 ms, respectively, while for actuators 2 and 3, the 10–90% rise times were 4.56 ms and 4.40 ms, respectively. The fall response time was also measured for each actuator, where it was found respectively as 5.12 ms and 4.88 ms for actuators 1 and 4 and 5.28 ms and 5.40 ms for actuators 2 and 3. The fall response time was higher than the rise response (20%, 14.8%, 16.67% and 15% higher for actuators 1, 4, 2 and 3, respectively). This difference is believed to be caused by the heat stored in the mirror plate flowing back into the actuators during the cooling phase [29].

The frequency response of the four actuators was measured and presented in Figure 7e. At a fixed amplitude of 1.5 V, the displacement of the laser beam (FOV) decreased with the frequency except for a few resonance peaks. Two main resonance peaks were detected for each actuator at 1400 and 1800 Hz with a secondary peak at 950 Hz. These peaks correspond respectively to the actuators tip–tilt modes and piston mode [28,29]. At high frequencies, the FOV decreased to reach 10% of its initial value at 10 Hz. We can also observe that in the frequency range between 1100 Hz and 1300 Hz, the FOV maintains a constant variation. In fact, the scanning response is a combination of thermal response and second-order mechanical response. The displacement of the mirror will decrease with increasing frequency, which is caused by the thermal response.

Finally, the impact of our scanning system on the pulse duration of the laser beam was investigated. Using an auto-correlator (Ape, mini) the auto-correlation trace of the laser pulse outgoing from the endoscopic double clad fiber (DCF) fiber was measured and is shown as the black curve in Figure 8. The auto correlation function was measured to be 70 fs, which corresponds to a pulse width of 45 fs for a pulse profile in sech^2^.

The auto-correlation trace of the laser pulse outgoing from the assembled scanning probe was also measured using an objective to collimate the laser beam toward the auto-correlator. Its duration was found to be 89 fs (the blue curve in Figure 8), corresponding to a pulse duration of 57 fs for a pulse profile in sech^2^. It is clear that the laser pulse duration did not greatly expand, suggesting that the scanning components did not alter the pulse duration, ensuring efficient two-photon excitation.

## 4. Discussion

Our main research project consists of developing a TPF multimodal endomicroscope dedicated to intraoperative brain imaging. This endomicroscope aims to improve the surgical act by delimitating the tumor infiltrating boundaries. It will be able to provide the neurosurgeon a fast and reliable multimodal quantitative and qualitative information on the tissue’s nature in order to establish a real-time diagnosis. This diagnosis information will be extracted through an already-established tissue optical database of several human brain tissue types that has been demonstrated in our past works.

In actuality, our developed endomicroscope is able to perform bimodal measurements of two photon fluorescence spectra and fluorescence lifetime with high efficiency [8]. Since the bimodality is validated, the next step on the track of reaching a multimodal endomicroscope is the scanning system probe development.

In this work, a homemade TPF scanning head probe dedicated to perform optical biopsy and intraoperative brain imaging when coupled with the developed endomicroscope was presented. The imaging probe head combines a miniaturized 2D scanning electrothermal MEMS mirror and micro-optical lenses in a two-lens configuration. This imaging probe is able to ensure, with a small outer diameter, a large imaging FOV as well as performing stable scanning acquisitions and delivers the shortest possible pulse duration to enhance the image contrast.

The characterization of the mechanical properties of the MEMS mirror starts by measuring the reflectivity of its aluminium-coated plate. It is an important parameter to be measured precisely in order to determine the signal transmission of the mirror and to quantify the excitation and collection efficiency on each wavelength since that the collected fluorescence signal is a polychromatic light. From this, several metabolic ratios could be derived from spectral fitting in order to provide important information about cellular metabolism [5] in different brain tissue types. These ratios are already established as discrimination factors between heathy and cancerous zones [5,6].

Moving to the actuation characteristics, and as mentioned before, a fast and stable scanning acquisition is required in order to obtain the best image quality. A quasi-linear OSA variation with the voltage drive signal was obtained for both Y and *X*-axis (Figure 5).

Indeed, Zemax simulations showed that a 560 µm FOV can be achieved when a ±10° OSA range is reached by the mirror. Our desired imaging FOV varied from 250 to 500 µm depending on the region of interest in the sample that the neurosurgeon wished to image. Therefore, and regarding the excellent linearity of the OSA variation, the maximum desired 500 µm FOV can be reached with ±8.9° OSA range for both axes which imposes the use of a very low operation voltage to drive the *Y*-axis actuators (1.84 V) and the *X*-axis actuators (2.69 V).

In addition, the OSA linearity was configured with a constant PDT variation through the two scanning axes. The standard variation of the obtained PDT values along a scanning axis was less than 10% (100 µs) around a calculated mean of 1000 µs, which means that our MEMS mirror actuators are able to perform a 90% stable scanning along the two scanning axes. This stability will allow the acquisition of the real representative images of the specimen and will avoid the blurry effects and the distortion of the acquired images.

The dynamic responses of the MEMS mirror actuators including the step response and frequency response were also characterized. Observed from the PDT variation curves, the step response time of each actuator, during the heating and cooling phase, was measured. This parameter is necessary to be known in order to adapt the drive and control system of the MEMS scanner to achieve a precise synchronization and trigger between the MEMS actuators and the photon counting detector. It will allow us to avoid the temporal phase shift between the start of the scan and the start of photon counting, which also aids to avoid the spatial shift that may occur between the pixel rows of an acquired image.

Factually, the response time of the actuators depends on the type of materials that compose the thermal bimorph actuator and their thermal expansion coefficients (TEC). The two materials that are widely used to fabricate the bimorph thermal actuators are Copper (Cu) with tungsten [29,30] and Aluminium (Al) with silicon dioxide (SiO2) [25,31]. Thus, due to their larger TEC difference (23.1 × 10^−6^/K for Al and 0.5 × 10^−6^/K for SiO2) comparing Cu/W actuators [30], the use of Al/SiO2 bimorph actuators was favored. In literature, the measured step response time of the widely used Cu/W Bimorph actuators was found at around 7 ms [29] and 15 ms [30], while the response time of our Al/SiO2 actuators was found much lower (4 ms for heating phase and 5 ms for cooling phase).

The frequency response was also investigated for each actuator, where the FOV showed a decreased behavior in function of the scanning frequency. This decrease limits the ability to change the scanning speed from a low to a fast acquisition without a FOV change. In the highest frequencies, the FOV reaches only 10% of its maximal value (Figure 7e) which imposes the increase of the used voltage in order to keep the same desired FOV.

This behavior, largely observed in the electrothermal actuators [14,15], can be surpassed through the drive and control system of the MEMS scanner by accounting for each frequency’s corresponding voltage value (amplitude of the control signal) in order to keep the same scanning FOV. However, the maximum advised driving voltage of an electrothermal actuator is around 5 V [14]. Surpassing this voltage overheats the Al/SiO2 layers and may therefore damage the actuators. For this reason, a value of 4 V was fixed as the maximum driving voltage to prevent the damage of the mirror actuators. This 4 V threshold will be applied and used to drive the fast scanning axis actuators only, since the slow axis is operated with a very low frequency in a raster pattern so no need to increase the voltage. Since the *Y*-axis actuators (1 and 4) showed a larger OSA value, they will be adopted to ensure the fast axis scanning of the desired raster pattern. Referring to the optical simulations and to the frequency response variation (Figure 7), the maximum operating frequency of the *Y*-axis as well as the maximum frame rate that can be reached for each desired imaging FOV was calculated and presented in Table 2. The maximum desired 500 µm FOV can be performed with a 1.8 fps rate, while the 2-fps rate can be reached with a 450 µm FOV.

The last essential parameter that we investigated was the effect of the probe optical components on the sub-fs pulse delivery. Our custom-built fiber-based endomicroscope is able to generate excitation pulses with 45 fs duration delivered to the distal end of a 5-meter length long microstructured double-clad photonic crystal fiber (DC-PCF). At the exit of our developed probe, this duration increased to reach 57 fs. These generated short pulses increase the laser peak power which will improve the generated fluorescence signal [7], and they are also critical to prevent photo-bleaching or photo-induced tissue damage within the focal volume. They are desired also for obtaining superior image quality, enhancing the imaging depth and for faster acquisition times [8].

## 5. Conclusions

In summary, a miniature two-photon fluorescence endomicroscopic probe head based on a 2-axis scanning electro-thermally actuated MEMS mirror with a total outer diameter of 4 mm was presented. This probe is dedicated to intraoperative brain imaging as well as performing optical biopsy. A combination of the most performing probe design and lens configuration with the most performing available electro-thermal actuators was adopted in order to obtain a well suitable tool for in-vivo biomedical imaging. The chosen probe elements (micro-optics and mirror) ensures obtaining a large scanning FOV with high imaging speed, resolution, packaging flexibility, small probe diameter and sub-fs pulse delivery.

As for actuation type and architecture, a small aperture mirror that included electro-thermal actuators that meet our needs was used for a fast and stable scanning acquisition. A high reflectivity of the mirror plate was measured, which ensures a good excitation and signal collection efficiency. A large and linear OSA range was reached at a low drive voltage for horizontal and vertical scan axes which allows for the reaching of a large FOV with a low power consumption. A 450 µm imaging FOV could be achieved with 2 fps as the imaging speed. A quasi-constant pixel dwell time was obtained along both scanning axes leading to a stable and a uniform scanning speed along a scanning pattern. In addition, the frequency response, the time response of the mirror actuators and the impact of the scanning system on the pulse duration were investigated.

Our ongoing work consists of developing an electronic acquisition system to control the developed device and to drive the MEMS mirror, synchronize the mirror with the detectors, perform photon counting and to quickly transfer the data for image reconstruction. Once the control software is finished, image acquisition will be performed and several parameters will be investigated.

This promising technology has the potential to give birth to a new generation of miniature TPF micro-endoscopic imaging probes that are capable of realizing in vivo early cancer detection as well as performing fast optical biopsy with high performance. MEMS characterization is a prerequisite step before implementing the first multimodal in vivo TPF micro-endoscopic imaging tool based on MEMS scanning micro-mirror technology.

## Figures and Tables

**Figure 1 micromachines-11-00704-f001:**
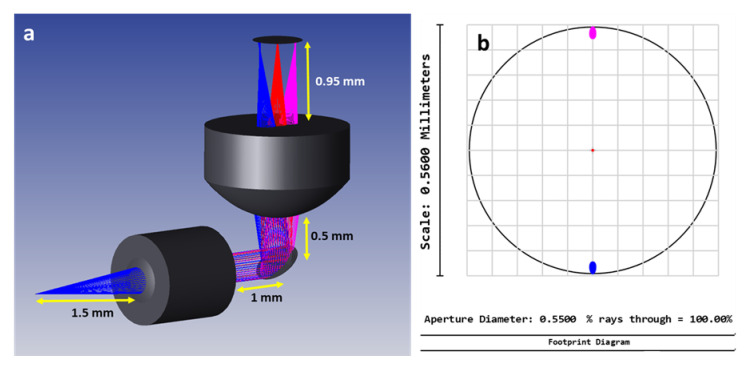
(**a**) Zemax shaded model of the probe design. (**b**) Footprint diagram of the tracked beam at the exit of the focusing lens.

**Figure 2 micromachines-11-00704-f002:**
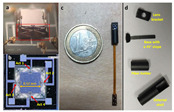
(**a**) An optical image of the Micro-Electro-Mechanical System (MEMS) mirror showing one of its two-level-ladder, double S-shaped actuators (red square); (**b**) Scanning electron microscopic (SEM) image showing the four actuators with the mirror plate; (**c**) Assembled probe; (**d**) Probe mechanical components.

**Figure 3 micromachines-11-00704-f003:**
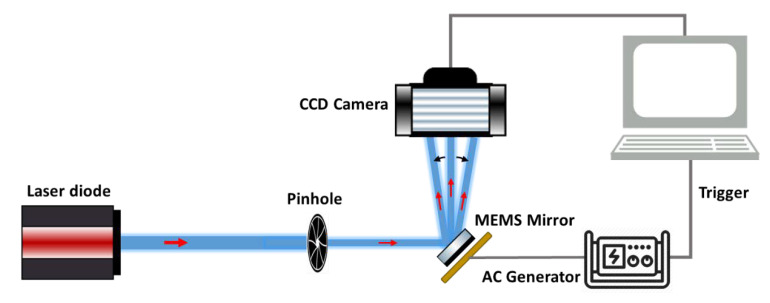
Schematic of the characterization setup of the mechanical properties of the MEMS actuators.

**Figure 4 micromachines-11-00704-f004:**
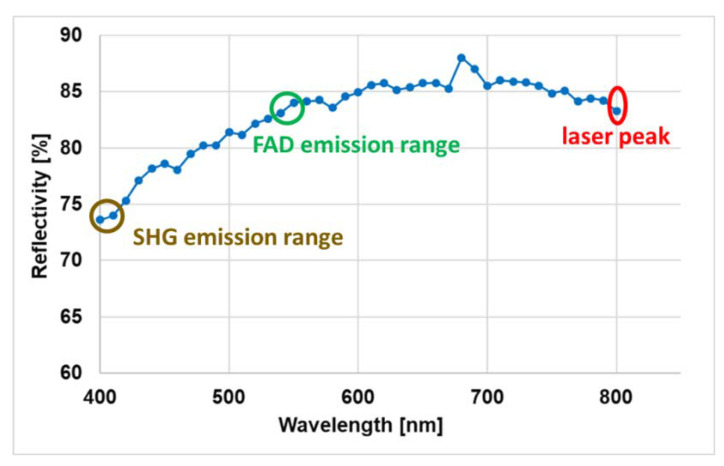
Measured reflectivity of the MEMS mirror plate.

**Figure 5 micromachines-11-00704-f005:**
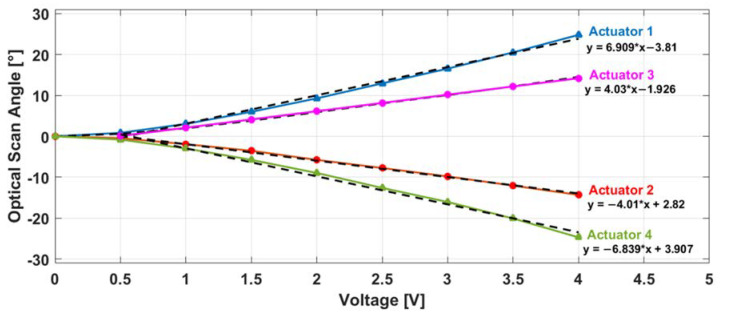
Optical scan angle variation with the driving voltage for the four actuators with the curve fitting equation of each one (black curves).

**Figure 6 micromachines-11-00704-f006:**
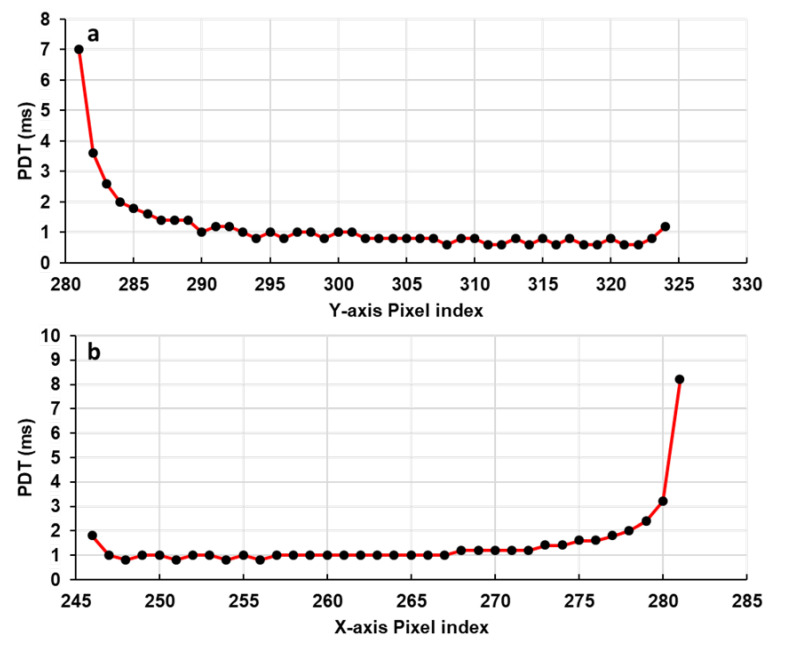
Measured pixel dwell time along a *Y*-axis scan line (**a**) and along an *X*-axis scan line (**b**).

**Figure 7 micromachines-11-00704-f007:**
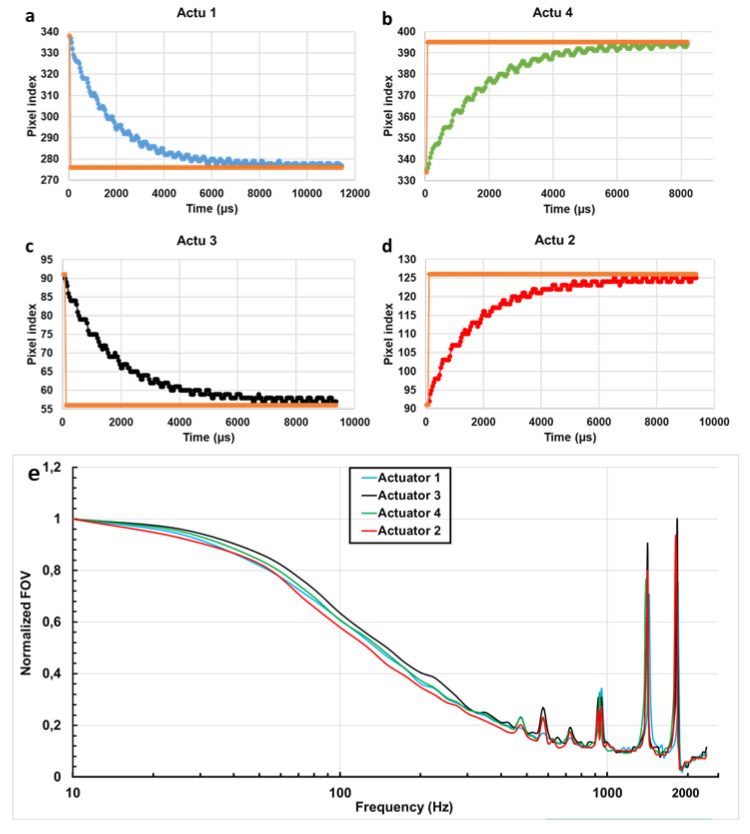
The dynamic responses of the MEMS mirror actuators. Step response of the MEMS mirror four actuators (**a**–**d**) in the rise phase (heating up). Frequency response of the four actuators (**e**) corresponding to the Normalized displacement of the beam (field of view, FOV) with the frequency variation at a fixed voltage of 1.5 V.

**Figure 8 micromachines-11-00704-f008:**
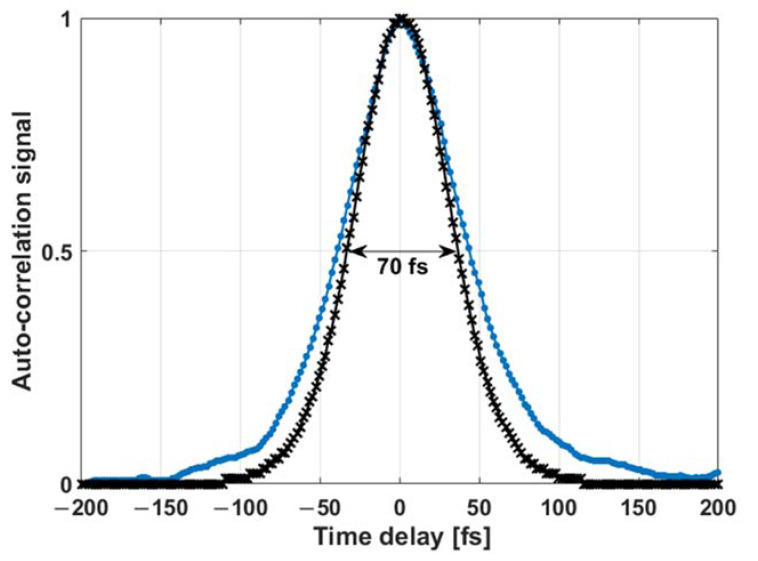
Autocorrelation trace at the output of the endoscopic double clad fiber (DCF) fiber (black curve) and at the output of the MEMS scanning system (blue curve).

**Table 1 micromachines-11-00704-t001:** Comparison of the main mechanical properties between three MEMS mirrors deploying Al/SiO2 dual S-shaped bimorph electrothermal actuators with a different actuators level architecture each.

	Single Level	Double Level	Three Level
OSA/Voltage	±13°/4 V [15]	±24°/4 V	±17.5°/5 V [26]
Response time	4.3–4.6 ms [25]	~4.5–5.5 ms	~28 ms [27]
First order resonant frequency	191 Hz [15]	950 Hz	1.3 KHz [26]

**Table 2 micromachines-11-00704-t002:** The operating frequencies for both scanning axes for each imaging FOV with the maximum frame rate that can be reached for each FOV.

FOV	Max Frequency *Y*-axis	Frequency *X*-Axis	Frame Rate
500 µm	450 Hz	0.9 Hz	1.8 fps
450 µm	500 Hz	1.11 Hz	2 fps
400 µm	550 Hz	1.37 Hz	2.2 fps
320 µm	775 Hz	2.42 Hz	3 fps
250 µm	1175 Hz	4.69 Hz	4.7 fps

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
