# Peer review of "A Customized Two Photon Fluorescence Imaging Probe Based on 2D scanning MEMS Mirror Including Electrothermal Two-Level-Ladder Dual S-Shaped Actuators"

_micromachines, 2020, doi:10.3390/mi11070704_

Round 1

Reviewer 1 Report

The manuscript by Mehidine et al. presents design and characterization of two photon miniature probe based on two photon fluorescence imaging for applications in in-vivo tumour imaging. I am not an expert in the field of fabrication of two photon probes for brain imaging. The article provides sufficient background information and details about the fabrication. It is not clear to me whether there will be any need to use any contrast agents or two photon active molecules which can bind to cancer cells specifically and so can be used to differentiate the cancer cells from normal cells. Authors can explain this and add some details in the manuscript. Since it is two photon absorption based is it that the technology is only suitable for near-surface organs or can this also be used to image the organs like kidney, heart and so on? Otherwise, mention this as limitations in the manuscript. No further review needed. The manuscript can be accepted ionce the authors address the issues mentioned above. 

Author Response

The reviewer brought up an important point that requires more clarification of our imaging methodology, which is based on label-free imaging with no use of any exogenous fluorophores or external dyes. This point was mentioned in the introduction in a not clear way, line 80-81, by citing that “Our team is developing a Two-Photon Fluorescence (TPF) intraoperative endomicroscope [17], able to efficiently excite four endogenous fluorophores”.

Changes were added in the introduction to more clarify our methodology, and to demonstrate the efficiency of using endogenous fluorescence for brain tissue discrimination by citing several past works that occurs in this context.

The second point that the reviewer has mentioned it is if this imaging technology, based on two-photon absorption, is suitable only for near-surface organs or it can be used to image other organs like kidney, heart … etc. In fact, our developed tool is dedicated for imaging during brain tumor surgery since the context of our research project is oriented toward brain tissue diagnosis. The issue of this context was detailed in the first section of the introduction, line 30-43. We added to the introduction, to define the context of the medical use of our developed tool.

Changes in the text, regarding the comments, are presented below and are marked in red in the manuscript:

Introduction: section from line 80 to 83 (ancient version of manuscript) was modified and moved to the section from line 51 to 66 (new version of manuscript).

Reviewer 2 Report

The submitted paper described MEMS mirror for 2D scanning in medical application, and investigated its operation properties.

(1) The MEMS mirror applied in this study seems to be known structure, and therefore the reviewer unfortunately can’t find the large advance from the conventional technologies.

(2) The authors should state the targeted specification of their probe system in the medical use. For examples, followings should be discussed.

(a) Scan frequency at both horizontal and vertical directions.

(b) Scan angle at both horizontal and vertical directions.

(c) Flatness of mirror surface.

(d) Degree of reflection.

(e) Size of probe system.

(f) Operation power (voltage).

(3) After the discussion of (2), the authors should cite and compare the present technologies. The reviewer thinks so many excellent MEMS mirrors have already developed up to now. For example, the group of Prof. Toshiyoshi in University Tokyo developed the excellent MEMS mirror for OCT probe application.

(4) Probe design

The authors should quantitatively compare and discuss the properties in the different lens configurations.

(5) MEMS mirror assembly

The authors should quantitatively show the alignment accuracy of the mechanical components at the packaging process.

(6) Results

The authors should discuss the following quantitatively.

#1) Linearity of scan angle against driving voltage.

#2) Uniformity among actuators.

#3) Repeatability of actuators.

#4) Position dependency of mirror device against the gravity dierction.

(7) Figure 7e.

The FOV value does not have the flat property against the frequency increment, thus t

he authors should discuss the dynamic frequency rage of the developed MEMS mirror (The maximum frequency in which the developed MEMS mirror can be used in the targeted application).

(8) The authors should show the optical scan properties of the developed probe system as shown in the title.

Author Response

(1) The MEMS mirror applied in this study seems to be known structure, and therefore the reviewer unfortunately can’t find the large advance from the conventional technologies.

We agree with the reviewer that the MEMS mirror structure used in our developed probe is a known structure, especially in OCT imaging for endoscopic applications where it was widely used by several groups in the world.

However, this structure is not widely used in TPF imaging probes, where most of the developed probes used the fiber scanning technique which includes a PZT ceramic tube. This point was mentioned in the introduction, line 83 to 85.

Indeed, the novelty in our developed probe and its mirror, comparing with the reported OCT imaging, is its actuators architecture (two-level ladder) and its combination with a two-lens configuration, contrary to the one lens configuration widely used in the most reported OCT imaging probes. The combination of this type of mirror (and its actuators architecture) with this two-lens configuration was not reported before for any TPF imaging probe. We have justified also in the text the choice of the actuation type, the actuators architecture as well as the lens configuration of the developed probe, in order to obtain the best possible characteristics for our specific medical use. Indeed, the table 1 added in materials and methods section, line 205-208 compares three different architectures of electrothermal Al/SiO2 actuators.

(2) The authors should state the targeted specification of their probe system in the medical use. For examples, followings should be discussed.

(a) Scan frequency at both horizontal and vertical directions.

(b) Scan angle at both horizontal and vertical directions.

(c) Flatness of mirror surface.

(d) Degree of reflection.

(e) Size of probe system.

(f) Operation power (voltage)

In this comments, the reviewer pointed out the lack of several aspects related to the frequency response and to the control parameters that were not well discussed in the manuscript. The Modifications were made and marked in red in the section below:

Materials and methods: line 214-215

Discussion line 340 to 346

Discussion line 367 to 372

Discussion line 394 to 415

Discussion line 403 to 407

Discussion: Table 2 was added

(3) After the discussion of (2), the authors should cite and compare the present technologies. The reviewer thinks so many excellent MEMS mirrors have already developed up to now. For example, the group of Prof. Toshiyoshi in University Tokyo developed the excellent MEMS mirror for OCT probe application.

We agree with the reviewer that several groups in the world developed excellent MEMS scanners, including the group of Prof. Toshiyoshi in University of Tokyo who develops endoscopic scanning probes dedicated for OCT endoscopic imaging and based on electrostatic MEMS mirrors. This type of actuation can be driven with a high operation speed and have a low power consumption. However, this type of actuation includes high driving voltage (~100V) and offers a lower scan range than the electrothermal. In the medical application of our team, and since the probe head will be applied directly in the patient head, a 100 V operation may occur risks for the patient in case of any error. Indeed, it is necessary to have a large scanning range in order to achieve a large imaging FOV [Jingjing Sun et al. doi: 10.1364/OE.18.012065].

As we mentioned in the introduction line 84-85, the most reported probes are addressed for OCT endoscopic imaging and includes a single-lens configuration which is not the most adapted for TPF imaging. Therefore, several comparisons between the used elements in our developed probe and the other types of elements in the literature were mentioned in the text, the added parts are marked in red in the sections below:

Line 172 to 177

Materials and methods: line 190 - 193

Materials and methods: line 201 – 208

Discussion: Line 387 to 389

(4) Probe design

The authors should quantitatively compare and discuss the properties in the different lens configurations

In literature, the group of Professor Zhongping Chen have conducted a study to compare the characteristics of three different lens configuration used for endoscopic optical imaging probe. We mentioned this study in the text as well as the main characteristics of each configuration, but we didn’t present quantitatively these characteristics. For that the related part of this point in the text was modified and marked in red in section 2 materials and methods, subsection 2.1 probe design, Line 122 to 165. Indeed, the main characteristics of each design can be briefly described through the figure below.

(5) MEMS mirror assembly

The authors should quantitatively show the alignment accuracy of the mechanical components at the packaging process.

The probe mechanical components, showed in figure 2.d, were designed using Solidworks software and manufactured by our collaborator in Florida university who provided the MEMS mirror and they were designed to ensure the maximum alignment accuracy between these elements and in a way that the packaging process don’t affect this alignment. The manufacturing process of these components, that hold up the optical lenses, as well as the manufacturing process of the MEMS mirror actuators was performed in the laboratory of our collaborator in Florida university. 

(6) Results

The authors should discuss the following quantitatively.

#1) Linearity of scan angle against driving voltage.

#2) Uniformity among actuators.

#3) Repeatability of actuators.

#4) Position dependency of mirror device against the gravity direction.

To respond this comment, and through the known information in our possession, several modifications were added to the discussion and results sections, where the linearity and the similarity between each scanning axis actuators OSA linearity was investigated and discussed by fitting the obtained curves. All changes were added in red in the text in the following sections:

Results: line 254 to 262

Discussion: line 386 to 391.

(7) Figure 7e.

The FOV value does not have the flat property against the frequency increment, thus the authors should discuss the dynamic frequency rage of the developed MEMS mirror (The maximum frequency in which the developed MEMS mirror can be used in the targeted application).

The non constant variation of the FOV against the frequency was discussed and we calculate the maximum frequency for each desired FOV. The added table 2 in the discussion section, line 413 to 414 mentioned these parameters for each FOV. Changes are marked in red in the manuscript in the sections below:

Results: line 314-318

Discussion: line 394 to 401

(8) The authors should show the optical scan properties of the developed probe system as shown in the title.

The main optical property that we are able to quantify in this stage of development is the delivered pulse duration through the optical components of the developed probe. This property was mentioned in figure 8 as well as in the results and discussion section:

Results: line 326 to 334

Discussion: line 415 to 422

Reviewer 3 Report

The manuscript is well written, with extensive literature review done and clear motivation behind the research. The section on  experimental setup was well furnished, with reasonable amount of characterization and discussion being made as well. The manuscript should be accepted in its present form.

Round 2

Reviewer 1 Report

Authors have made changes in the manuscript to account for the suggestions/comments made by the reviewers. The current version of the manuscript can be accepted for publication.